# Cilostazol effectiveness in reducing drug-coated stent restenosis in the superficial femoral artery: The ZERO study

**Takashi Miura**[1,2]*, **Yusuke Miyashita**[2,3], **Koji Hozawa**[4], **Tatsuki Doijiri**[5], **Tamon Kato**[2], **Naoki Hayakawa**[6], **Naoto Hashizume**[7], **Masatsugu Nakano**[8], **Uichi Ikeda**[1,2], **Koichiro Kuwahara**[2], on behalf of the ZERO Investigators[¶]

1 Department of Cardiology, Nagano Municipal Hospital, Nagano, Japan, 2 Department of Cardiovascular Medicine, Shinshu University Hospital, Matsumoto, Japan, 3 Department of Cardiology, Nagano Red Cross Hospital, Nagano, Japan, 4 Department of Cardiology, New Tokyo Hospital, Matsudo, Japan, 5 Department of Cardiology, Yamato Seiwa Hospital, Yamato, Japan, 6 Department of Cardiology, Asahi General Hospital, Asahi, Japan, 7 Department of Cardiology, Shinonoi General Hospital, Nagano, Japan, 8 Department of Cardiology, Tokyo General Hospital, Nakano, Japan

¶ Membership of the ZERO Investigators is listed in the Acknowledgments.
* miuramen10miuramen@yahoo.co.jp

**Data Availability Statement:** All relevant data are within the paper and its Supporting Information files.

## Abstract

### Purpose

Drug-eluting stents (DESs) play an important role in endovascular therapy (EVT) for femoro-popliteal (FP) lesions. Cilostazol improves patency after bare-metal nitinol stent (BNS) implantation for femoropopliteal lesions. This study aimed to establish whether cilostazol is effective in improving the patency of DESs and determine whether BNS or DESs with or without cilostazol are more effective in improving the 12-month patency after EVT for FP lesions.

### Materials and methods

In this prospective, open-label, multicenter study, 85 patients with symptomatic peripheral artery disease due to de novo FP lesions were enrolled and treated with DESs with cilostazol from eight cardiovascular centers between April 2018 and May 2019. They were compared with 255 patients from the DEBATE SFA study, in which patients were randomly assigned to the BNS, BNS with cilostazol, or DES groups. The primary endpoint was the 12-month patency rate using duplex ultrasound (peak systolic velocity ratio < 2.5). This study was approved by the ethics committee of each hospital.

### Results

The 12-month patency rates for the BNS, BNS with cilostazol, DES, and DES with cilostazol groups were 77.6%, 93.1%, 82.8%, and 94.2%, respectively (p = 0.007). The 12-month patency rate was higher in the DES with cilostazol group than in the DES group (p = 0.044). In small vessels, the DES with cilostazol group had a higher patency rate than the DES group (100.0% vs. 83.4%, p = 0.023).

**Funding:** The author(s) received no specific funding for this work.

**Competing interests:** The authors have declared that no competing interests exist.

## Conclusions

DES with cilostazol showed better patency than DES alone. Cilostazol improved patency after EVT with DES in FP lesions and small vessels.

## Clinical trial registration

University Hospital Medical Information Network Clinical Trials Registry (no. UMIN 000032473).

## Introduction

Drug-eluting stents (DESs) play an important role in endovascular therapy (EVT) for femoro-popliteal (FP) lesions. Advances in pharmaceutical technologies have improved the primary patency rate after EVT for FP lesions [1–4]. A recent trial comparing two well-established DESs showed a primary patency rate of 81%–86% at 12 months [5].

The effects of cilostazol in reducing the in-stent restenosis (ISR) rate of bare-metal nitinol stents (BNSs) for FP lesions are well recognized. Cilostazol reduces the ISR rate by nearly half in first-generation BNS and by 20% in second-generation BNS [6, 7]. Furthermore, post hoc analyses of the ZEPHYR study showed that cilostazol was strongly associated with a lower ISR rate within one year after DES implantation for FP lesions.

However, no prospective study has yet determined the efficacy of cilostazol in reducing the ISR rate of DESs for FP lesions. Furthermore, it is not known whether BNS or DES with cilostazol are more effective in reducing ISR after EVT for FP lesions. The Zilver PTX with cilostazol in the Superficial Femoral Artery (ZERO) trial was designed to address this gap in knowledge and to clarify 1) whether cilostazol is effective in reducing ISR of DES, and 2) which strategy is more effective in reducing the one-year ISR rate after EVT for FP.

## Materials and methods

### Study design

This was a prospective, open-label, multicenter study involving 85 patients with symptomatic peripheral artery disease due to *de novo* FP lesions enrolled from eight cardiovascular centers which participated DEBATE in SFA study between April 2018 and May 2019. Participants underwent stenting with a Zilver PTX stent and were treated with cilostazol (DES with cilostazol group). These patients were compared with 255 patients from the drug-eluting versus bare-metal stent implantation with or without cilostazol in the treatment of the superficial femoral artery (DEBATE SFA) study, as historical control data, in which between March 2014 and April 2016, 255 patients with symptomatic PAD due to de novo FP lesions were enrolled from 25 cardiovascular centers and randomly assigned to the BNS group (Misago stent implantation without cilostazol), BNS with cilostazol group (Misago stent implantation with cilostazol), or DES group (Zilver PTX stent implantation without cilostazol).

Written informed consent was obtained from all patients. This study was approved by the ethics committees of each participating hospital (Nagano municipal hospital 0033/2018.4.12; Shinshu university hospital 4186/2018.10.2; Nagano red cross hospital 155/2018.11.12; New Tokyo Hospital 0153-1/2018.5.28; Yamato seiwa hospital 00012/2018.6.4; Asahi general hospital 201851510/2018.5.15; Shinonoi general hospital H30-5/2020.12.21; Tokyo general hospital 64/2018.7.10) and was registered in University Hospital Medical Information Network Clinical Trials Registry, and was performed in accordance with the Declaration of Helsinki and its later amendments.

## Participants

Patients with symptomatic claudication or rest pain (Rutherford 2–4) and *de novo* FP lesions were included. Patients with dual antiplatelet therapy after coronary DES implantation, severe heart failure precluding the use of cilostazol, FP lesion with inflow aortoiliac lesions, poor below-the-knee runoff (defined as < 1 below-the-knee runoff), anticoagulant treatment, bleeding tendency, or acute or sub-acute limb ischemia were excluded. These criteria were the same as those used in the DEBATE SFA study.

## Post-procedural medical treatment

The enrolled patients (DES with cilostazol group) were treated with aspirin (100 mg/day) and cilostazol (200 mg/day) for 12 months and with clopidogrel (75 mg/day) for two months after EVT. In the DEBATE SFA study, the BNS group was treated with aspirin (100 mg/day) for 12 months and clopidogrel (75 mg/day) for one month, the BNS with cilostazol group was treated with aspirin (100 mg/day) and cilostazol (200 mg/day) for 12 months, and the PTX group was treated with aspirin (100 mg/day) and clopidogrel (75 mg/day) for 12 months after EVT.

## Endovascular procedure and lesion measurement

The decision on the approach site was left at the discretion of the physicians. In nearly all cases, a 6-French sheath was inserted into the femoral artery via the contralateral or ipsilateral approach. After 50–100 IU/kg of heparin infusion, the lesion was crossed with a 0.014- or 0.035-inch guidewire. The lesion was dilated with a semicompliant or noncompliant scoring balloon. The recommended stent size was 1 mm greater than the diameter of the distal vessel before stent implantation. Post-dilatation was performed using a noncompliant balloon with a size equivalent to the diameter of the distal vessel. The decision to use intravascular ultrasonography was made at the physician's discretion.

The proximal and distal reference vessels and lesion length were measured from angiographic data.

## Outcome assessment

Outcomes were assessed over a 12-month period. The primary outcome measure was the restenosis rate, which was determined by a peak systolic velocity ratio (PSV) $\geq 2.5$ (we changed definition of PSV from >2.0 to $\geq 2.5$ because that of almost all trial was $\geq 2.5$ [8]), derived from duplex ultrasonography performed at the 1-, 3-, 6-, and 12-months follow-up visits. The secondary outcome measure was the occurrence of major adverse limb events (MALE), which was determined as a composite of limb-related death, target lesion revascularization (TLR), major amputation, major bleeding, and definite or probable stent thrombosis. In ISR patients, the Tosaka classification was applied to the different groups.

## Statistical analysis

The sample size was estimated based on the binary restenosis rates from previous trial. An overall sample size of 90 patients was expected to have 80% power to detect a difference in 1-year ISR after Zilver PTX stenting with cilostazol compared with that without cilostazol at a 2-sided $\alpha$ of 0.05, assuming a binary restenosis rate of 30% in the Zilver PTX with cilostazol group and 50% in the Ziver PTX without cilostazol group. Finally, the sample size was determined 90 patients which is same as that of DEBATE SFA study. All analyses were based on the intention-to-treat principle. For baseline characteristics, continuous variables are presented as mean ± standard deviation or median (interquartile range) or percentages for dichotomous

variables and were compared using the Student *t*-tests and chi-squared tests (asymptotic or Fisher's exact test) for categorical variables. Statistical significance was set at p < 0.05. Primary end point were estimated at 12 months using the Kaplan-Meier method, and p-values were calculated using the log-rank test. Secondary endpoints were and estimated at 12 months using Cox proportional hazards regression models. Statistical analyses were performed using SPSS version 24 (IBM Corp., Armonk, NY, USA).

## Results

Among the 85 patients, three without indication for EVT were excluded. A total of 82 patients in the DES with cilostazol group were compared with the DES (n = 85), BNS with cilostazol (n = 85), and BNS groups (n = 85) (Fig 1). Baseline and lesion characteristics were well matched among the four groups (Tables 1 and 2). Lesion characteristics across the four study groups showed many similarities: the median length of the treated segment was nearly 100 mm, 41% of the lesions were chronic total occlusion lesions, 37% of the lesions were Trans-Atlantic Inter-Society Consensus II (TASC II) C/D lesions, and 54% were found in small vessels (< 5 mm diameter). In addition, the frequency of involvement of the proximal superficial femoral artery (SFA) or popliteal artery and the distal reference vessel size were comparable among the four groups. Five (6.1%) patients died during the 12-month follow-up period in the DES with cilostazol group (1, 5, and 5 patients died in the BNS, BNS with cilostazol, and DES groups, respectively). Cilostazol treatment was continued for all patients during the 12-month follow-up period. Significant differences were found in the 12-month patency rate among the DES with cilostazol, DES, BNS with cilostazol, and BNS groups (94.2% vs. 82.8% vs. 93.1% vs. 77.6%, respectively, p = 0.007) (Fig 2).

### Comparison of the patency rate between the DES and DES with cilostazol groups

Overall, the 12-month patency rate was remarkably higher in the DES with cilostazol group than in the DES group (94.2% vs. 82.8%, p = 0.044) (Fig 3A). Furthermore, Cox proportional hazards

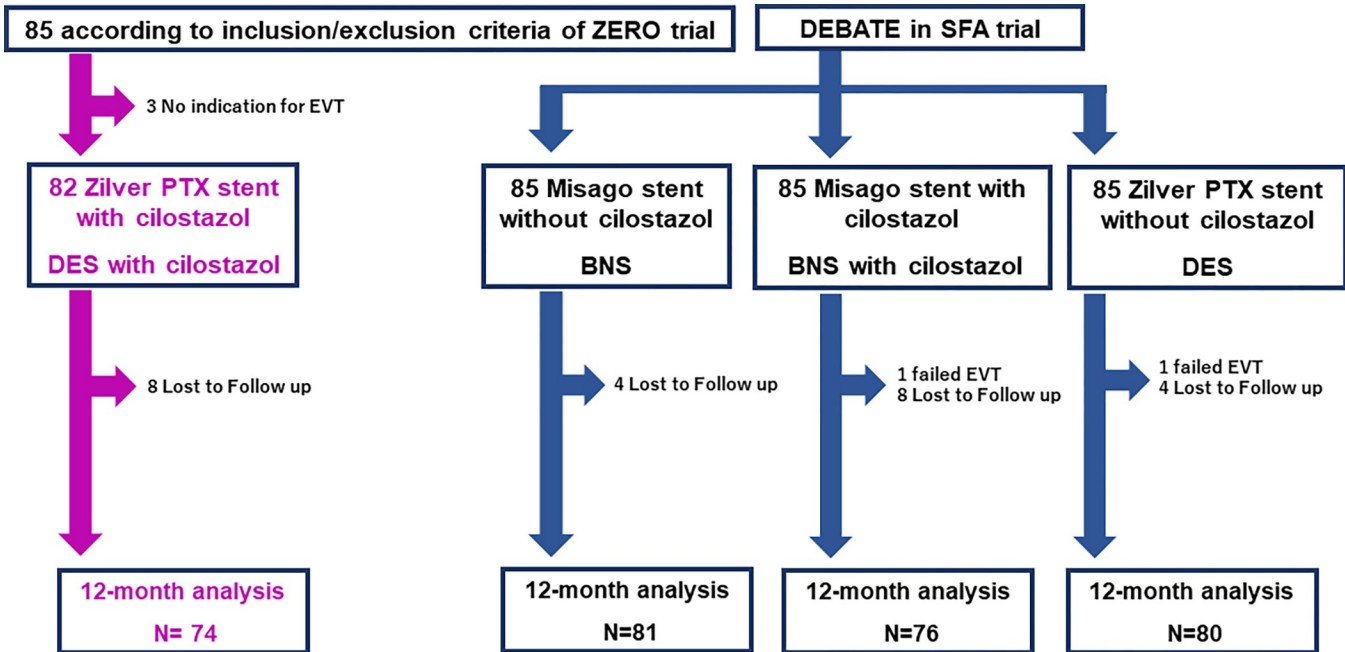

**Fig 1. Study flowchart.** Abbreviations: BNS, bare nitinol stent; DES, drug-eluting stent; EVT, endovascular therapy.

**Table 1. Baseline characteristics.**

| Variables | BNS (n = 85) | BNS with cilostazol (n = 85) | DES (n = 85) | DES with cilostazol (n = 82) |
|---|---|---|---|---|
| Age (years) | 73.4 ± 8.0 | 72.7 ± 9.1 | 73.1 ± 7.8 | 74.2 ± 9.7 |
| Male (%) | 55 (64.7) | 57 (67.0) | 60 (70.6) | 52 (63.4) |
| Body mass index (kg/m$^2$) | 21.9 [19.7, 24.4] | 21.6 [20.0, 23.8] | 22.3 [20.1, 24.5] | 23.1 [20.2, 24.8] |
| Hypertension (%) | 73 (85.9) | 69 (81.2) | 68 (80.0) | 64 (78.0) |
| Dyslipidemia (%) | 49 (57.7) | 52 (61.2) | 52 (61.2) | 49 (59.8) |
| Diabetes mellitus (%) | 48 (56.5) | 42 (49.4) | 50 (58.8) | 42 (51.2) |
| Insulin use (%) | 13 (15.3) | 16 (18.8) | 16 (18.8) | 15 (18.3) |
| Current smoker (%) | 21 (24.7) | 20 (23.5) | 24 (28.2) | 23 (28.1) |
| Previous smoker (%) | 43 (50.6) | 45 (52.9) | 45 (52.9) | 43 (52.4) |
| eGFR (mL/min/1.73 m$^2$) | 48.3 ± 28.5 | 49.3 ± 29.1 | 52.1 ± 28.1 | 43.4 ± 23.8 |
| Hemodialysis (%) | 19 (22.4) | 17 (20.0) | 17 (20.0) | 15 (18.3) |
| Previous stroke (%) | 20 (23.5) | 15 (17.7) | 19 (22.4) | 11 (13.4) |
| CAD (%) | 34 (40.0) | 32 (37.7) | 44 (51.8) | 38 (46.3) |
| Previous heart failure (%) | 1 (1.2) | 6 (7.1) | 6 (7.1) | 12 (14.6) |
| LV dysfunction (%) | 3 (3.5) | 2 (2.4) | 5 (5.9) | 8 (9.8) |
| Previous bleeding (%) | 0 (0.0) | 3 (3.5) | 4 (4.7) | 3 (3.7) |
| Rutherford class III (%) | 32 (37.6) | 26 (30.6) | 33 (38.8) | 37 (45.1) |
| Rutherford class IV (%) | 10 (11.8) | 14 (16.5) | 6 (7.1) | 20 (24.4) |
| ABI at enrollment | 0.65 [0.57, 0.75] | 0.66 [0.58, 0.74] | 0.68 [0.58, 0.77] | 0.66 [0.56, 0.77] |
| Medication on discharge | | | | |
| Statins (%) | 53 (62.4) | 43 (50.6) | 54 (63.5) | 40 (48.8) |
| ACE I/ARBs (%) | 55 (64.7) | 44 (51.8) | 45 (52.9) | 43 (52.4) |
| β-blockers (%) | 16 (18.8) | 25 (29.4) | 29 (34.1) | 23 (28.1) |
| Ca-antagonists (%) | 53 (62.4) | 44 (51.8) | 45 (52.9) | 49 (59.8) |
| Procedure | | | | |
| Crossover (%) | 44 (51.8) | 45 (52.9) | 46 (54.1) | 33 (40.2) |
| Ipsilateral (%) | 41 (48.2) | 39 (45.9) | 37 (43.5) | 41 (50.0) |
| IVUS usage (%) | 66 (77.7) | 52 (61.2) | 59 (69.4) | 59 (72.0) |
| No. of stent implantations | | | | |
| 1 (%) | 51 (60.0) | 47 (55.3) | 43 (50.6) | 54 (65.9) |
| 2 (%) | 27 (31.8) | 31 (36.5) | 25 (29.4) | 15 (18.3) |
| 3 (%) | 6 (7.1) | 6 (7.1) | 15 (17.7) | 5 (6.1) |
| 4 (%) | 1 (1.2) | 0 (0.0) | 1 (1.2) | 0 (0.0) |

Abbreviations: BNS, bare-metal nitinol stents; DES, drug-eluting stents; eGFR, estimated glomerular filtration rate; CAD, coronary artery disease; LV, left ventricular; ABI, ankle-brachial index; ACE-I, angiotensin-converting-enzyme inhibitor; ARB, angiotensin II receptor blocker; IVUS, intravascular ultrasonography.

Data are shown as the mean ± SD, median [interquartile range], or n (percentages).

regression models showed that cilostazol is independent predictor of reducing ISR in patients with DES adjusted by proximal diameter of the reference vessel and the below knee runoffs which factors are significantly difference between groups (HR 0.26, 95%CI 0.07–0.93; P = 0.038). MALE, all-cause death, TLR, and major bleeding were comparable between the two groups (Table 3).

## Comparison of the ISR rate between the DES with cilostazol and BNS with cilostazol groups

The 12-month patency rates between the DES with cilostazol and BNS with cilostazol groups was comparable. Furthermore, MALE, all-cause death, TLR, and major bleeding were

**Table 2. Lesion characteristics.**

| Variables | BNS (n = 85) | BNS with cilostazol (n = 85) | DES (n = 85) | DES with cilostazol (n = 82) | p-value |
|---|---|---|---|---|---|
| TASC IIC/D (%) | 33 (38.8) | 35 (41.2) | 32 (37.7) | 25 (30.5) | 0.76 |
| CTO (%) | 31 (36.5) | 33 (38.8) | 43 (50.6) | 31 (37.8) | 0.45 |
| Calcification (%) | 49 (57.7) | 50 (58.8) | 49 (57.7) | 33 (40.2) | 0.29 |
| Lesion length (mm) | 96.0 [44.9, 184.0] | 101.3 [45.6, 210.0] | 110.5 [49.3, 249.0] | 99.0 [58.8, 160.0] | 0.72 |
| Proximal diameter of the reference vessel (mm) | 5.1 ± 1.1 | 5.3 ± 1.2 | 5.3 ± 1.0 | 5.5 ± 1.1 | 0.03 |
| Distal diameter of the reference vessel (mm) | 4.9 ± 0.9 | 5.2 ± 1.0 | 4.9 ± 0.8 | 5.0 ± 0.9 | 0.25 |
| P1 involvement (%) | 15 (17.7) | 15 (17.7) | 15 (17.7) | 23 (28.1) | 0.18 |
| Proximal SFA involvement (%) | 19 (22.4) | 22 (25.9) | 20 (23.5) | 13 (15.8) | 0.64 |
| Number of below the knee runoffs, 1/2/3 | 26/39/20 | 23/42/18 | 19/37/27 | 35/26/13 | 0.034 |

Abbreviations: BNS, bare-metal nitinol stents; DES, drug-eluting stents; CTO, chronic total occlusion; P1, popliteal 1; SFA, superficial femoral artery; TASC, Trans-Atlantic Inter-Society Consensus.

Data are shown as the mean ± SD, median [interquartile range], n (percentage), or number per group (n/n/n).

comparable between the two groups. Subgroup analysis revealed a comparable patency rate between the two groups for patients with TASC II C/D lesions (100.0% vs. 93.5%, p = 0.347) and small vessels (< 5 mm) (100.0% vs. 97.6%, p = 0.392).

## Comparison of ISR types

ISR occurred in three patients, and importantly, all cases were classified as Tosaka class I in the DES with cilostazol group. The distribution of the ISR type according to the patient group was as follows: Tosaka class I occurred in 10 (44%; BNS), 8 (89%; BNS with cilostazol), and 10 (62%; DES) patients; Tosaka class II occurred in 7 (30%; BNS), 0 (0%; BNS with cilostazol), and 3 (19%; DES) patients; Tosaka class III occurred in 6 (26%; BNS), 1 (11%; BNS with cilostazol), and 3 (19%; DES) patients. The distribution of the ISR type was significantly different among the four groups (p = 0.002) (Fig 4).

## Discussion

The ZERO study is a prospective, open-label, multicenter study designed to evaluate the patency rate of DES with cilostazol after EVT for FP lesions. Data from the DEBATE SFA study were used to compare the patency rates of DES with cilostazol and BNS, BNS with cilostazol, and DES after EVT for FP lesions.

To the best of the authors' knowledge, the ZERO study is the first to reveal that (1) cilostazol significantly improves the DES patency rate at 12 months, (2) all patients with ISR had class I (focal) type in the DES with cilostazol group, and (3) in patients with small vessels and TASC C/D FP lesions, there was no ISR in the DES with cilostazol group at the 12-month follow-up.

Soga et al. first revealed that cilostazol significantly reduced the restenosis rate after EVT with stenting or ballooning alone for FP lesions in a prospective randomized trial [9]. Iida et al. showed that cilostazol reduced the 1-year ISR for EVT with first-generation BNS implantation for FP lesions (BNS vs. BNS with cilostazol, 49% vs. 20%, P = 0.0001). Subanalysis of the ZEPHYR study revealed that the one-year ISR rate after Zilver PTX implantation for FP lesions was significantly lower in the cilostazol group than in the non-cilostazol group by propensity score matching analysis (33% vs. 51%, P = 0.008). Patients in the ZEPHYR study had a mean lesion length of nearly 170 mm and mean reference vessel diameter of 5.0 mm; 37% had chronic total occlusions, 19% had stent restenosis lesions, 31% were on dialysis, and 29% had critical limb ischemia [10]. This included severe cases, and the patients' baseline characteristics

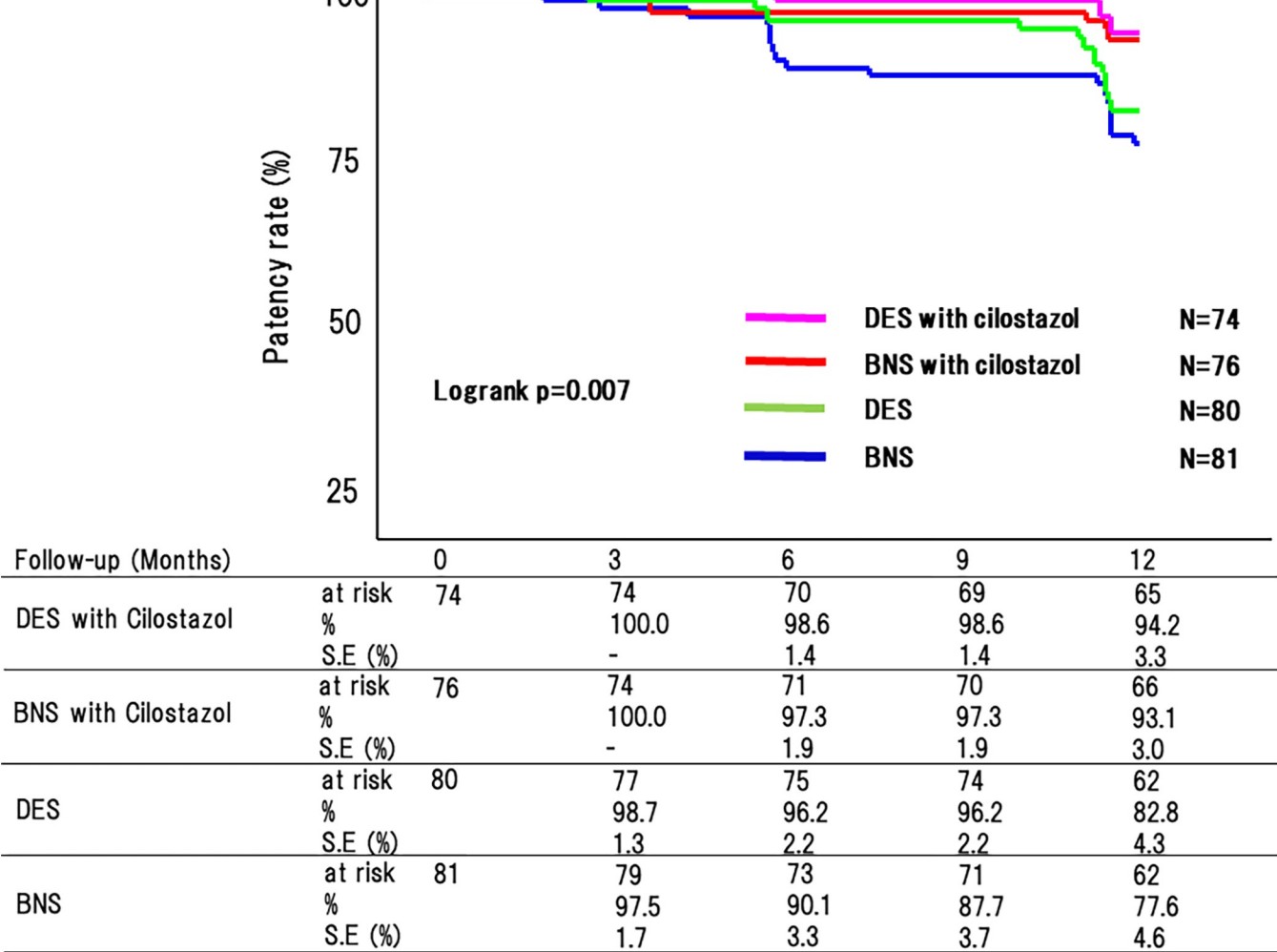

**Fig 2. Overall primary patency in all groups.** The 12-month patency rate differed significantly among the four groups (DES with cilostazol: 94.2% vs. BNS with cilostazol 93.1% vs. DES without cilostazol 82.8%, BNS without cilostazol 77.6%, p = 0.0007). Abbreviations: DES, drug-eluting stent; BNS, bare nitinol stent; DES, drug-eluting stent; S.E; standard error.

were heterogeneous. Therefore, the ZERO study is the first to prospectively evaluate the effectiveness of reducing the ISR of cilostazol after EVT with DES for *de novo* FP lesions.

## Cilostazol is effective in reducing ISR for small vessels and long lesions

In patients with FP lesions, small vessels and long lesions remain the limitations of EVT. Currently, drug-coated balloons (DCBs) are the first choice for treating small vessels. In a retrospective analysis, Kamioka et al. revealed that the three-year patency rate of balloon angioplasty was superior to that of BNS stenting for FP lesions [11]. Moreover, DCBs could significantly reduce restenosis compared with balloon angioplasty in FP lesions, particularly in diabetic or female patients [12]. These findings suggest that DCB use may be a preferable strategy for reducing restenosis in small vessels. However, in cases of flow-limiting dissection after balloon dilatation, a stent is required. The ZEPHYR trial showed that DES is not suitable for small-vessel FP lesions [13]. In the present study, although DES with cilostazol was significantly superior to DES in reducing restenosis in small vessels, there was a similar patency rate

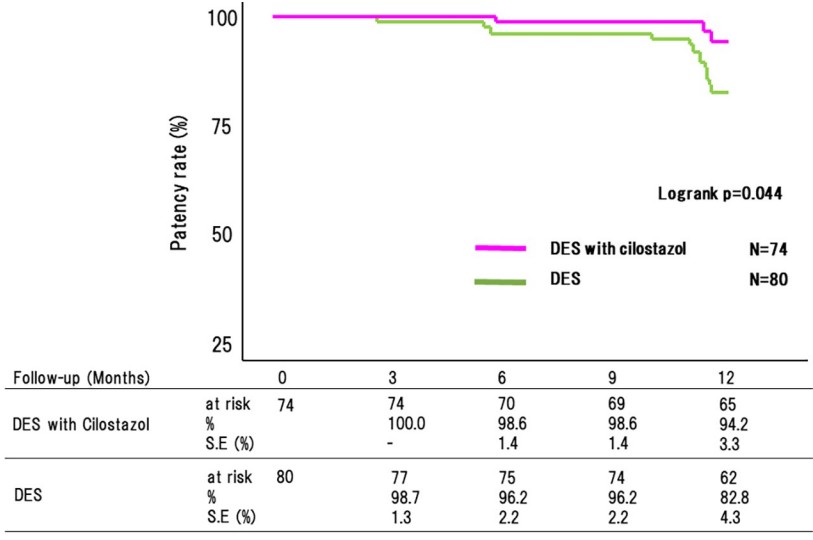

| Follow-up (Months) | | 0 | 3 | 6 | 9 | 12 |
|---|---|---|---|---|---|---|
| DES with Cilostazol | at risk | 74 | 74 | 70 | 69 | 65 |
| | % | | 100.0 | 98.6 | 98.6 | 94.2 |
| | S.E (%) | | - | 1.4 | 1.4 | 3.3 |
| DES | at risk | 80 | 77 | 75 | 74 | 62 |
| | % | | 98.7 | 96.2 | 96.2 | 82.8 |
| | S.E (%) | | 1.3 | 2.2 | 2.2 | 4.3 |

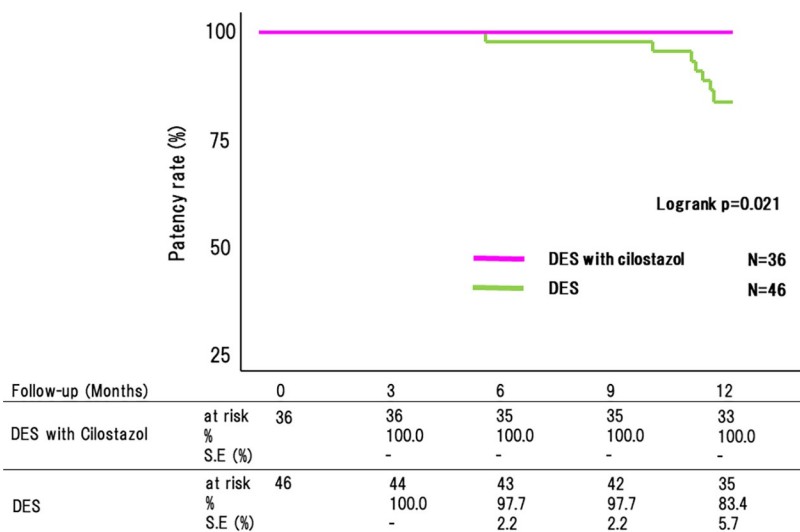

| Follow-up (Months) | | 0 | 3 | 6 | 9 | 12 |
|---|---|---|---|---|---|---|
| DES with Cilostazol | at risk | 36 | 36 | 35 | 35 | 33 |
| | % | | 100.0 | 100.0 | 100.0 | 100.0 |
| | S.E (%) | | - | - | - | - |
| DES | at risk | 46 | 44 | 43 | 42 | 35 |
| | % | | 100.0 | 97.7 | 97.7 | 83.4 |
| | S.E (%) | | - | 2.2 | 2.2 | 5.7 |

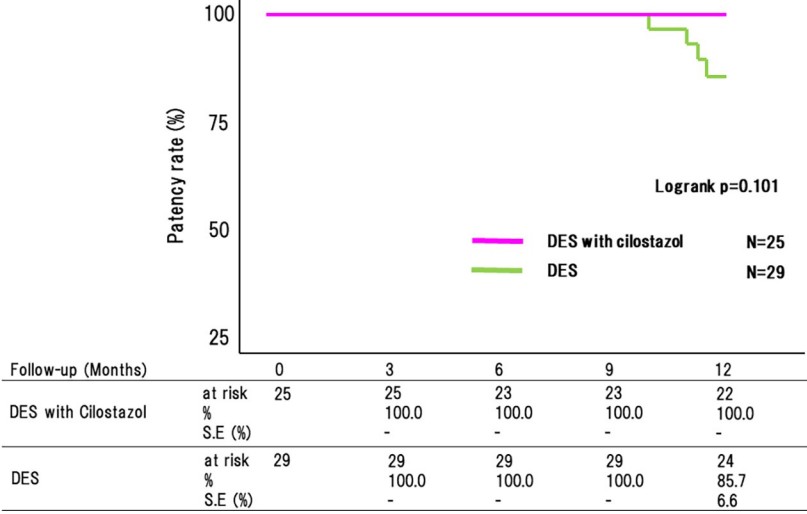

| Follow-up (Months) | | 0 | 3 | 6 | 9 | 12 |
|---|---|---|---|---|---|---|
| DES with Cilostazol | at risk | 25 | 25 | 23 | 23 | 22 |
| | % | | 100.0 | 100.0 | 100.0 | 100.0 |
| | S.E (%) | | - | - | - | - |
| DES | at risk | 29 | 29 | 29 | 29 | 24 |
| | % | | 100.0 | 100.0 | 100.0 | 85.7 |
| | S.E (%) | | - | - | - | 6.6 |

**Fig 3. Patency rate difference between the DES with and without cilostazol groups.** (A) Overall data. The 12-month patency rate was significantly higher in the DES with cilostazol group than in the DES group (94.2% vs. 82.8%, p = 0.044). (B) Patients with small-vessel lesions. The patency rate was significantly higher in the DES with cilostazol group than in the DES group (100.0% vs. 83.4%, p = 0.021). (C) Patients with TASC II C/D lesions. The DES with cilostazol group had a higher patency rate than the DES group, although this difference was not significant (100.0% vs. 85.7%, p = 0.101). Abbreviations: DES, drug-eluting stent; BNS, bare nitinol stent; TASC II, Trans-Atlantic Inter-Society Consensus II; S.E; standard error.

between the DES with cilostazol and BNS with cilostazol groups. Therefore, the authors recommend using cilostazol when using metal stents for small-vessel FP lesions, irrespective of whether these are DES or BNS.

Longer lesions decrease the linear patency rate of BNS and DES after EVT for FP lesions [13, 14]. Zeller et al. reported that a heparin-bonded stent graft (Viabahn) was suitable for long lesions based on a one-year primary patency rate of 67% after EVT [15]. However, the present study revealed that the one-year primary patency rate after EVT with DES with cilostazol for long lesions was 100.0%, which is comparable to that of BNS with cilostazol. Based on the present data, cilostazol could maintain a high patency rate for TASC II C/D lesions treated with BNS as well as DES.

## ISR pattern

The ISR distribution by type has been reported as follows: Tosaka class I, II, and II patterns were found in 29%, 38%, and 33% of cases following BNS use, respectively [16], while class I, II, and III patterns were found in 50%, 25%, and 25% of cases following DES use, respectively [17].

Furthermore, the DEBATE SFA trial showed that cilostazol reduced class II and III ISR by 0% and 11%, respectively [7]. The present study revealed that DESs with cilostazol were not associated with class II or III ISR. Type III ISR occurred more frequently, particularly with re-occlusion after balloon angioplasty. This may indicate that DES with cilostazol is effective in reducing ISR and improving the treatment of FP by EVT.

**Table 3. Event rate at 1 year.**

| | DES+C (n = 82) | BNS (n = 85) | BNS+C (n = 85) | DES (n = 84) | DES+C vs. DES | | DES+C vs. BNS+C | | DES+C vs. BNS | |
|---|---|---|---|---|---|---|---|---|---|---|
| | | | | | HR (95% CI) | p-value | HR (95% CI) | p-value | HR (95% CI) | p-value |
| Male (%) | 3 (3.7) | 14 (16.6) | 5 (6.5) | 5 (6.3) | 0.89 (0.20–3.98) | 0.875 | 0.79 (0.39–1.62) | 0.521 | 0.67 (0.44–1.02) | 0.063 |
| All-cause death (%) | 5 (6.1) | 1 (1.2) | 6 (7.1) | 5 (6.0) | 1.13 (0.33–3.92) | 0.844 | 0.94 (0.52–1.70) | 0.839 | 1.84 (0.90–3.76) | 0097 |
| Death due to limb (%) | 0 (0.0) | 0 (0.0) | 1 (1.2) | 0 (0.0) | | | | | | |
| Major bleeding (%) | 1 (1.2) | 4 (4.8) | 1 (1.2) | 2 (2.4) | 0.54 (0.05–5.97) | 0.616 | 1.02 (0.26–4.08) | 0.977 | 0.65 (0.32–1.36) | 0.253 |
| TLR (%) | 3 (3.7) | 8 (9.7) | 4 (5.1) | 3 (3.6) | 0.81 (0.14–4.90) | 0.822 | 0.73 (0.31–1.70) | 0.458 | 0.65 (0.39–1.10) | 0.106 |
| Major amputation (%) | 0 (0.0) | 0 (0.0) | 0 (0.0) | 0 (0.0) | | | | | | |

Participants in groups DES+C, BNS, BNS+C, and DES were administered DES with cilostazol, BNS, BNS with cilostazol, and DES treatment, respectively.

Abbreviations: DES, drug-eluting stents; BNS, bare-metal nitinol stents; HR, hazard ratio; MALE, major adverse limb event, defined as a composite of limb-related death, target lesion revascularization, major amputation, and major bleeding; TLR, target lesion revascularization.

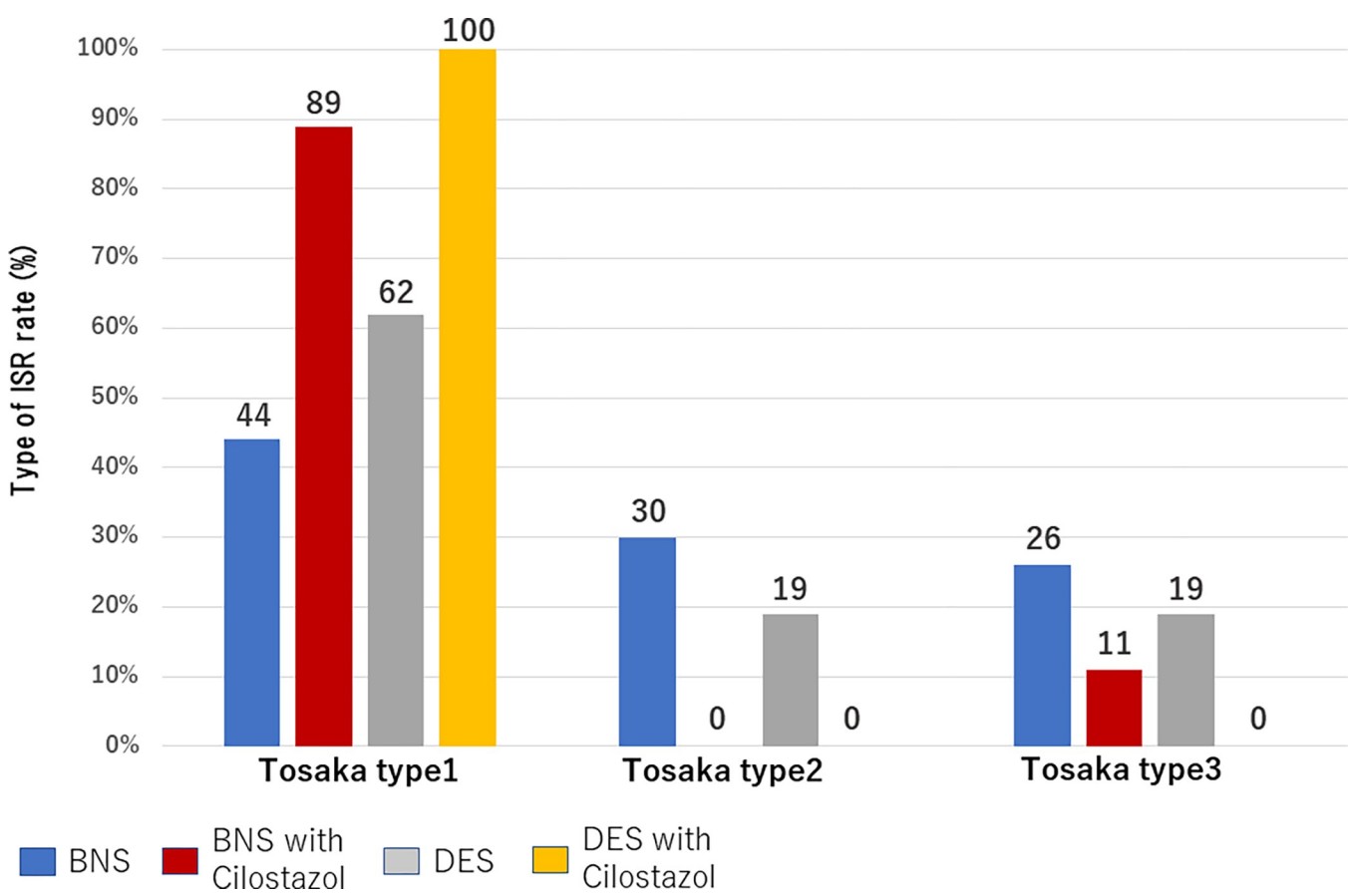

**Fig 4. ISR pattern of each group.** The DES with cilostazol group had no type II or III ISR compared with the other three groups. Abbreviations: ISR, in-stent restenosis, DES, drug-eluting stent.

### Possibility of cilostazol's effect

Recently, paclitaxel-eluting stents (Eluvia) have demonstrated superior patency rate outcomes to polymer-free paclitaxel-coated stents (Zilver PTX) [5]. Although this study revealed an extremely high patency rate for Zilver PTX with cilostazol in FP lesions, cilostazol treatment combined with Eluvia can possibly achieve improved results compared with Zilver PTX with cilostazol.

### Limitations

Our study was not a randomized controlled trial, although the authors compared the present data with historical data and used the same enrollment and exclusion criteria. Statistical matching technique was not performed due to small number of enroll patients. Furthermore, adjusted analysis was not enough due to small number of endpoints. Although the authors had no independent core laboratory to assess the initial procedure, our colleagues evaluated all angiographic data. Finally, stent fracture was one of the reasons for ISR, but the authors were unable to collect these data comprehensively and could not describe the stent fracture.

### Conclusions

The patency rate was greater in patients in whom DES with cilostazol was used than in patients in whom only DES was used. Cilostazol improved the patency rate after EVT with DES for FP lesions and small vessels.

## Supporting information

**S1 Checklist.**
(PDF)

**S1 File.**
(DOC)

**S1 Protocol.**
(DOC)

## Acknowledgments

We thank all the ZERO investigators, especially Minako Aono.

## Author Contributions

**Data curation:** Tatsuki Doijiri, Naoki Hayakawa.

**Formal analysis:** Tamon Kato.

**Investigation:** Takashi Miura.

**Validation:** Masatsugu Nakano.

**Writing – original draft:** Takashi Miura.

**Writing – review & editing:** Yusuke Miyashita, Koji Hozawa, Naoto Hashizume, Uichi Ikeda, Koichiro Kuwahara.

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
