## [Decision Letter · Decision Letter 0]

10 Mar 2022

PONE-D-21-22546

Cilostazol Effectiveness in Reducing Drug-Coated Stent Restenosis in the Superficial Femoral Artery: The ZERO Study

PLOS ONE

Dear Dr. Miura,

Thank you for submitting your manuscript to PLOS ONE. After careful consideration, we feel that it has merit but does not fully meet PLOS ONE’s publication criteria as it currently stands. Therefore, we invite you to submit a revised version of the manuscript that addresses the points raised during the review process.

We look forward to receiving your revised manuscript.

Kind regards,

Salvatore De Rosa

Academic Editor

PLOS ONE

2. Please report the date(s) in which you received ethics approval for your study.

3. We note that your clinical trial registration does not include sufficient information. Please update your record to include all relevant sections, such as study design and intervention.

“NO”

“NO authors have competing interests”

6. Please upload a copy of Figures 4 and 5, to which you refer in manuscript. If the figure is no longer to be included as part of the submission please remove all reference to it within the text.

8. We noticed you have some minor occurrence of overlapping text with the following previous publication, which needs to be addressed:

- https://www.ahajournals.org/doi/10.1161/CIRCINTERVENTIONS.118.006564

In your revision ensure you cite all your sources (including your own works), and quote or rephrase any duplicated text outside the methods section. Further consideration is dependent on these concerns being addressed.

Reviewers' comments:

Reviewer's Responses to Questions

**Comments to the Author**

1. Is the manuscript technically sound, and do the data support the conclusions?

Reviewer #1: Partly

Reviewer #2: Partly

2. Has the statistical analysis been performed appropriately and rigorously? 

Reviewer #1: No

Reviewer #2: I Don't Know

3. Have the authors made all data underlying the findings in their manuscript fully available?

Reviewer #1: Yes

Reviewer #2: Yes

4. Is the manuscript presented in an intelligible fashion and written in standard English?

Reviewer #1: Yes

Reviewer #2: Yes

5. Review Comments to the Author

Reviewer #1: General comments:

The authors present analyses where data from an open-label study were pooled with data from a historical randomized trial.

Overall, I think these analyses have merit, though more information is needed to understand why pooling these two trials is reasonable. I believe more needs to be done with the analytic methods as well. Pooling of studies can create bias since data were collected at different times and, potentially, different places. These biases need to be addressed, or at least fully discussed to inform readers, and for me that has not been done here.

Specific comments:

1. (p.4) I believe that you mean that the data…*will be* made available.

2. (p.5) A really key part of these analyses is how comparable the DES+cilostazol patients are to the DEBATE SFA study participants. One part that it completely glossed over are the locations that people are coming from. Were the eight centers from DES+cilostazol ones that were included in DEBATE SFA? If not, are they from similar geographic areas? These sorts of questions should be answered somewhere in the manuscript.

3. (p.7) Please include further information on the sample size. For instance, how was this powered, i.e., using a particular test? What values were assumed, e.g., what effect size?

4. (p.7; Table 3) I think these analyses should be run with proportional hazards regression models (aka Cox models). There is a categorical exposure here so, best that I understand, you are running a bunch of pairwise log-rank tests. That is probably resulting in a higher type I error rate than expected.

5. In addition, the Cox model will allow you to control for other potential confounders which I think needs to be done here. Some of the baseline characteristics in table 1 suggest that including them in a model might have some impact. I am not sure whether any covariates should go into a final model, but this should at least be tried, especially given that these are pooled data.

6. Finally, in the Cox model, I think clustering by center should be accounted for in the model, probably with some sort of marginal model, e.g., via the sandwich estimator. This could also be done with a frailty.

7. You might also consider performing some sensitivity analyses in regards to the pooling, especially in terms of the centers. It's hard for me to know what to recommend here information is lacking. Maybe you could limit the data from DEBATE SFA to those that were in DES+cilostazol. Or maybe some sort of intelligent matching of centers or participants.

8. (Table 1) Significance testing for baseline imbalance in randomized trials been has regarded as unnecessary and potentially misleading (see Altman DG. Comparability of randomised groups. The Statistician 1985; 125-136; Senn, S. Testing for baseline balance in clinical trials. Statistics in Medicine 1994; 1715-1726). I recommend removing the significance testing entirely from table 1.

9. There's no page 15 and then numbering starts on page 16. (It would be nice to have the line numbers throughout the document in future submissions.)

10. (Table 3) I think this table would be easier if the arm names could be used instead of "Group 1" etc.

11. (Figure 2) These survival curves should have confidence bands on them to depict the variability in these estimates. Though, four curves with bands will make this pretty busy. You may need to break these out into panels.

12. (Figure 3) Same comment as figure 2, though with just two curves, panels shouldn't be needed.

Reviewer #2: I read with interest the manuscript #PONE-D-21-22546 by Dr Miura and colleagues entitled “Cilostazol Effectiveness in Reducing Drug-Coated Stent Restenosis in the Superficial Femoral Artery: The ZERO Study”.

This is a prospective, open-label, multicenter study evaluating the efficacy of cilostazol in improving the patency of DESs and determine whether BNS or DESs with or without cilostazol are more effective in improving the 12-month patency after EVT for FP lesions.

The manuscript is well written and the topic is of interest.

However, the manuscript presents several issues:

- In the “outcome assessments” sub-section, concerning the peak systolic velocity (PSV) ratio cut-off used for ISR determination, authors state “we changed definition of PSV from >2.0 to > 2.5 because that of almost all trial was > 2.5”. First, authors should at least cite the studies they are referring to. Second and most important, in the DEBATE in SFA trial, it seems that the primary end point was evaluated using a peak systolic velocity ratio <2.0, as stated in the “Outcomes Assessment” section of that study1. Since the PSV was used to determine restenosis, the primary outcome of this study, this represents a major issue of the manuscript. Please Clarify.

- When comparing cohorts with historical control data, statistical methods including a propensity score matching would improve the quality of the analysis.

- In the “Results” section, authors state that “no significant differences were found in the 12-month patency rate among the DES with cilostazol, DES, BNS with cilostazol, and BNS groups (94.2% vs. 82.8% vs. 93.1% vs. 77.6%, respectively, p = 0.007) (Figure 2)”; however, in the Figure 2 description is reported that “the 12-month patency rate differed significantly among the four groups (DES with cilostazol: 94.2% vs. BNS with cilostazol 93.1% vs. DES without cilostazol 82.8%, BNS without cilostazol 77.6%, p = 0.0007) with different p value. Please clarify.

References:

1. Miura T, Miyashita Y, Soga Y, Hozawa K, Doijiri T, Ikeda U, et al. Drug-eluting versus bare-metal stent implantation with or without cilostazol in the treatment of the superficial femoral artery. Circ Cardiovasc Interv. 2018;11(8): e006564. doi: 10.1161/CIRCINTERVENTIONS.118.006564.

6. PLOS authors have the option to publish the peer review history of their article (what does this mean?). If published, this will include your full peer review and any attached files.

Reviewer #1: No

Reviewer #2: No

---

## [Author Response · Author response to Decision Letter 0]

25 Apr 2022

We wish to express our appreciation to the Reviewer for their comments, which have helped us significantly improve the paper. 

Reviewer #1: General comments:

The authors present analyses where data from an open-label study were pooled with data from a historical randomized trial.

Overall, I think these analyses have merit, though more information is needed to understand why pooling these two trials is reasonable. I believe more needs to be done with the analytic methods as well. Pooling of studies can create bias since data were collected at different times and, potentially, different places. These biases need to be addressed, or at least fully discussed to inform readers, and for me that has not been done here.

Specific comments:

1. (p.4) I believe that you mean that the data…*will be* made available.

Response: We appreciate the Reviewer`s comment on this point. We deleted the sentence.

2. (p.5) A really key part of these analyses is how comparable the DES+cilostazol patients are to the DEBATE SFA study participants. One part that it completely glossed over are the locations that people are coming from. Were the eight centers from DES+cilostazol ones that were included in DEBATE SFA? If not, are they from similar geographic areas? These sorts of questions should be answered somewhere in the manuscript.

Response: We thank the Reviewer`s comment on this point. The eight centers participated DEBATE in SFA study As Reviewer's comment, we described that in Materials and Methods on P 5. 

3. (p.7) Please include further information on the sample size. For instance, how was this powered, i.e., using a particular test? What values were assumed, e.g., what effect size?

Response: We appreciate the Reviewer`s comment on this point. As Reviewer's comment, we described further information on the sample size in Statistical analysis on P 7. 

4. (p.7; Table 3) I think these analyses should be run with proportional hazards regression models (aka Cox models). There is a categorical exposure here so, best that I understand, you are running a bunch of pairwise log-rank tests. That is probably resulting in a higher type I error rate than expected.

Response: We appreciate the Reviewer`s comment on this point. As Reviewer's comment, we reanalysed secondary endpoints by Cox proportional hazards regression models.

5. In addition, the Cox model will allow you to control for other potential confounders which I think needs to be done here. Some of the baseline characteristics in table 1 suggest that including them in a model might have some impact. I am not sure whether any covariates should go into a final model, but this should at least be tried, especially given that these are pooled data.

6. Finally, in the Cox model, I think clustering by center should be accounted for in the model, probably with some sort of marginal model, e.g., via the sandwich estimator. This could also be done with a frailty.

Response: We appreciate the Reviewer`s comment on this point. As Reviewer's comment, we used Cox model to adjust other factors, especially which factors have significant difference between groups. However, because number of endpoints are not so much, we could adjust only a few predictors. We described them in Limitation on P23. Unfortunately we didn`t have the data of frailty.

7. You might also consider performing some sensitivity analyses in regards to the pooling, especially in terms of the centers. It's hard for me to know what to recommend here information is lacking. Maybe you could limit the data from DEBATE SFA to those that were in DES+cilostazol. Or maybe some sort of intelligent matching of centers or participants.

Response: We thank the Reviewer`s comment on this point. All participating centers of current study participated DEBATE in SFA study As Reviewer's comment, we described that in Materials and Methods on P 5. 

8. (Table 1) Significance testing for baseline imbalance in randomized trials been has regarded as unnecessary and potentially misleading (see Altman DG. Comparability of randomised groups. The Statistician 1985; 125-136; Senn, S. Testing for baseline balance in clinical trials. Statistics in Medicine 1994; 1715-1726). I recommend removing the significance testing entirely from table 1.

Response: We appreciate the Reviewer`s comment on this point. As Reviewer's comment, we removed significance testing.

9. There's no page 15 and then numbering starts on page 16. (It would be nice to have the line numbers throughout the document in future submissions.)

Response: We appreciate the Reviewer`s comment on this point. As Reviewer's comment, we removed that page.

10. (Table 3) I think this table would be easier if the arm names could be used instead of "Group 1" etc.

Response: We appreciate the Reviewer`s comment on this point. As Reviewer's comment, we changed the arm names form <Group1,,,> to <DES+C,,,>.

11. (Figure 2) These survival curves should have confidence bands on them to depict the variability in these estimates. Though, four curves with bands will make this pretty busy. You may need to break these out into panels.

Response: We really appreciate the Reviewer`s comment on this point. As Reviewer's comment, four curves with bands will make busy, and still busy to describe 95%CI outside of graph. Thus, we added standard error outside of survival curves.

12. (Figure 3) Same comment as figure 2, though with just two curves, panels shouldn't be needed.

Response: We really appreciate the Reviewer`s comment on this point. As Reviewer's comment, Figure 3 was just two curves. But they were close each other. Thus, we added standard error outside of survival curves.

Reviewer #2: I read with interest the manuscript #PONE-D-21-22546 by Dr Miura and colleagues entitled “Cilostazol Effectiveness in Reducing Drug-Coated Stent Restenosis in the Superficial Femoral Artery: The ZERO Study”.

This is a prospective, open-label, multicenter study evaluating the efficacy of cilostazol in improving the patency of DESs and determine whether BNS or DESs with or without cilostazol are more effective in improving the 12-month patency after EVT for FP lesions.

The manuscript is well written and the topic is of interest.

However, the manuscript presents several issues:

- In the “outcome assessments” sub-section, concerning the peak systolic velocity (PSV) ratio cut-off used for ISR determination, authors state “we changed definition of PSV from >2.0 to > 2.5 because that of almost all trial was > 2.5”. First, authors should at least cite the studies they are referring to. 

Response: We thank the Reviewer`s comment on this point. As Reviewer's comment, we added the paper in which there were definition of PSV ≥ 2.5 into reference (we noticed our mistake and corrected from > to ≥ ). 

Second and most important, in the DEBATE in SFA trial, it seems that the primary end point was evaluated using a peak systolic velocity ratio <2.0, as stated in the “Outcomes Assessment” section of that study1. Since the PSV was used to determine restenosis, the primary outcome of this study, this represents a major issue of the manuscript. Please Clarify.

Response: We appreciate the Reviewer`s comment on this point. As Reviewer's comment, definition of ISR in the DEBATE in SFA was PSV > 2.0. Thus, in this study, we re-analysed ISR rate with PSV ≥2.5 in DEBATE in SFA data with this study`s data. 

- When comparing cohorts with historical control data, statistical methods including a propensity score matching would improve the quality of the analysis.

Response: We appreciate the Reviewer`s comment on this point. We completely agree with Reviewer's comment. Actually, we tried to analyse propensity score matching, but we coludn`t, because number of original enroll patients were small. We described in Limitation on P 23.

- In the “Results” section, authors state that “no significant differences were found in the 12-month patency rate among the DES with cilostazol, DES, BNS with cilostazol, and BNS groups (94.2% vs. 82.8% vs. 93.1% vs. 77.6%, respectively, p = 0.007) (Figure 2)”; however, in the Figure 2 description is reported that “the 12-month patency rate differed significantly among the four groups (DES with cilostazol: 94.2% vs. BNS with cilostazol 93.1% vs. DES without cilostazol 82.8%, BNS without cilostazol 77.6%, p = 0.0007) with different p value. Please clarify.

Response: We really appreciate the Reviewer`s comment on this point. As Reviewer's comment, we corrected from No significant… to Significant…. in Results on P 8.

---

## [Decision Letter · Decision Letter 1]

22 Jun 2022

Cilostazol Effectiveness in Reducing Drug-Coated Stent Restenosis in the Superficial Femoral Artery: The ZERO Study

PONE-D-21-22546R1

Dear Dr. Miura,

We’re pleased to inform you that your manuscript has been judged scientifically suitable for publication and will be formally accepted for publication once it meets all outstanding technical requirements.

Kind regards,

Salvatore De Rosa

Academic Editor

PLOS ONE

Additional Editor Comments (optional):

Reviewers' comments:

Reviewer's Responses to Questions

**Comments to the Author**

1. If the authors have adequately addressed your comments raised in a previous round of review and you feel that this manuscript is now acceptable for publication, you may indicate that here to bypass the “Comments to the Author” section, enter your conflict of interest statement in the “Confidential to Editor” section, and submit your "Accept" recommendation.

Reviewer #1: All comments have been addressed

Reviewer #2: All comments have been addressed

2. Is the manuscript technically sound, and do the data support the conclusions?

Reviewer #1: (No Response)

Reviewer #2: Partly

3. Has the statistical analysis been performed appropriately and rigorously? 

Reviewer #1: (No Response)

Reviewer #2: Yes

4. Have the authors made all data underlying the findings in their manuscript fully available?

Reviewer #1: (No Response)

Reviewer #2: Yes

5. Is the manuscript presented in an intelligible fashion and written in standard English?

Reviewer #1: (No Response)

Reviewer #2: Yes

6. Review Comments to the Author

Reviewer #1: (No Response)

Reviewer #2: The authors edited their manuscript addressing the majority of the reviwers' comments, improving the overall quality.

7. PLOS authors have the option to publish the peer review history of their article (what does this mean?). If published, this will include your full peer review and any attached files.

Reviewer #1: No

Reviewer #2: No

---

## [Editor Report · Acceptance letter]

27 Jun 2022

PONE-D-21-22546R1 

Cilostazol Effectiveness in Reducing Drug-Coated Stent Restenosis in the Superficial Femoral Artery: The ZERO Study 

Dear Dr. Miura:

I'm pleased to inform you that your manuscript has been deemed suitable for publication in PLOS ONE. Congratulations! Your manuscript is now with our production department. 

Kind regards, 

on behalf of

Dr. Salvatore De Rosa 

Academic Editor

PLOS ONE